# PUTTING THEORY TO WORK: FROM LEARNING BOUNDS TO META-LEARNING ALGORITHMS

## ABSTRACT

Most of existing deep learning models rely on excessive amounts of labeled training data in order to achieve state-of-the-art results, even though these data can be hard or costly to get in practice. One attractive alternative is to learn with little supervision, commonly referred to as few-shot learning (FSL), and, in particular, *meta-learning* that learns to learn with few data from related tasks. Despite the practical success of meta-learning, many of its algorithmic solutions proposed in the literature are based on sound intuitions, but lack a solid theoretical analysis of the expected performance on the test task. In this paper, we review the recent advances in meta-learning theory and show how they can be used in practice both to better understand the behavior of popular meta-learning algorithms and to improve their generalization capacity. This latter is achieved by integrating the theoretical assumptions ensuring efficient meta-learning in the form of regularization terms into several popular meta-learning algorithms for which we provide a large study of their behavior on classic few-shot classification benchmarks. To the best of our knowledge, this is the first contribution that puts the most recent learning bounds of meta-learning theory into practice for the task of few-shot classification.

## 1 INTRODUCTION

Since the very seeding of the machine learning field, its algorithmic advances were inevitably followed or preceded by the accompanying theoretical analyses establishing the conditions required for the corresponding algorithms to learn well. Such a synergy between theory and practice is reflected in numerous concepts and learning strategies that took their origins in the statistical learning theory: for instance, the famous regularized risk minimization approach is directly related to the minimization of the complexity of the hypothesis space, as suggested by the generalization bounds established for supervised learning (Vapnik, 1992), while most of the adversarial algorithms in transfer learning (*e.g.*, DANN from (Ganin & Lempitsky, 2015)) follow the theoretical insights provided by the seminal theory of its domain (Ben-David et al., 2010).

Even though many machine learning methods now enjoy a solid theoretical justification, some more recent advances in the field are still in their preliminary state which requires the hypotheses put forward by the theoretical studies to be implemented and verified in practice. One such notable example is the emerging field of *meta-learning*, also called *learning to learn* (LTL), where the goal is to produce a model on data coming from a set of (meta-train) source tasks to use it as a starting point for learning successfully a new previously unseen (meta-test) target task with little supervision. This kind of approach comes in particularly handy when training deep learning models as their performance crucially depends on the amount of training data that can be difficult and/or expensive to get in some applications. Several theoretical studies (Baxter, 2000; Pentina & Lampert, 2014; Maurer et al., 2016; Amit & Meir, 2018; Yin et al., 2020)[1] provided probabilistic meta-learning bounds that require the amount of data in the meta-train source task *and* the number of meta-train tasks to tend to infinity for efficient meta-learning. While capturing the underlying general intuition, these bounds do not suggest that all the source data is useful in such learning setup due to the

---

[1]We omit other works for meta-learning via online convex optimization (Finn et al., 2019; Balcan et al., 2019; Khodak et al., 2019; Denevi et al., 2019) as they concern a different learning setup.

additive relationship between the two terms mentioned above. To tackle this drawback, two very recent studies (Du et al., 2020; Tripuraneni et al., 2020) aimed at finding deterministic assumptions that lead to faster learning rates allowing meta-learning algorithms to benefit from all the source data. Contrary to probabilistic bounds that have been used to derive novel learning strategies for meta-learning algorithms (Amit & Meir, 2018; Yin et al., 2020), there was no attempt to verify the validity of the assumptions leading to the fastest known learning rates in practice or to enforce them through an appropriate optimization procedure.

In this paper, we bridge the meta-learning theory with practice by harvesting the theoretical results from Tripuraneni et al. (2020) and Du et al. (2020), and by showing how they can be implemented algorithmically and integrated, when needed, to popular existing meta-learning algorithms used for few-shot classification (FSC). This latter task consists in classifying new data having seen only few training examples, and represents one of the most prominent examples where meta-learning has shown to be highly efficient. More precisely, our contributions are three-fold:

1. We identify two common assumptions from the theoretical works on meta-learning and show how they can be verified and forced via a novel regularization scheme.

2. We investigate whether these assumptions are satisfied for popular meta-learning algorithms and observe that some of them naturally satisfy them, while others do not.

3. With the proposed regularization strategy, we show that enforcing the assumptions to be valid in practice leads to better generalization of the considered algorithms.

The rest of the paper is organized as follows. After presenting preliminary knowledge on the meta-learning problem in Section 2, we detail the existing meta-learning theoretical results with their corresponding assumptions and show how they can be enforced via a general regularization technique in Section 3. Then, we provide an experimental evaluation of several popular few-shot learning (FSL) methods in Section 4 and highlight the different advantages brought by the proposed regularization in practice. Finally, we conclude and outline future research perspectives in Section 5.

## 2 PRELIMINARY KNOWLEDGE

We start by formally defining the meta-learning problem following the model described in Du et al. (2020). To this end, we assume having access to $T$ source tasks characterized by their respective data generating distributions $\{\mu_t\}_{t=1}^T$ supported over the joint input-output space $\mathbb{X} \times \mathbb{Y}$ with $\mathbb{X} \subseteq \mathbb{R}^d$ and $\mathbb{Y} \subseteq \mathbb{R}$. We further assume that these distributions are observed only through finite size samples of size $n_1$ grouped into matrices $\mathbf{X}_t = (\mathbf{x}_{t,1}, \ldots, \mathbf{x}_{t,n_1}) \in \mathbb{R}^{n_1 \times d}$ and vectors of outputs $y_t = (y_{t,1}, \ldots, y_{t,n_1}) \in \mathbb{R}^{n_1}, \forall t \in [[T]] := \{1, \ldots, T\}$.

Given this set of tasks, our goal is to learn a shared representation $\phi$ belonging to a certain class of functions $\Phi := \{\phi \mid \phi : \mathbb{X} \to \mathbb{V}, \mathbb{V} \subseteq \mathbb{R}^k\}$ and linear predictors $\mathbf{w}_t \in \mathbb{R}^k, \forall t \in [[T]]$ grouped in a matrix $\mathbf{W} \in \mathbb{R}^{T \times k}$. More formally, this is done by solving the following optimization problem:

$$\hat{\phi}, \widehat{\mathbf{W}} = \underset{\phi \in \Phi, \mathbf{W} \in \mathbb{R}^{T \times k}}{\arg\min} \frac{1}{2Tn_1} \sum_{t=1}^T \sum_{i=1}^{n_1} \ell(y_{t,i}, \langle \mathbf{w}_t, \phi(\mathbf{x}_{t,i}) \rangle), \tag{1}$$

where $\ell : \mathbb{Y} \times \mathbb{Y} \to \mathbb{R}_+$ is a loss function. Once such a representation is learned, we want to apply it to a new previously unseen target task observed through a pair $(\mathbf{X}_{T+1} \in \mathbb{R}^{n_2 \times d}, y_{T+1} \in \mathbb{R}^{n_2})$ containing $n_2$ samples generated by the distribution $\mu_{T+1}$. We expect that a linear classifier $\mathbf{w}$ learned on top of the obtained representation leads to a low true risk over the whole distribution $\mu_{T+1}$. More precisely, we first use $\hat{\phi}$ to solve the following problem:

$$\hat{\mathbf{w}}_{T+1} = \underset{\mathbf{w} \in \mathbb{R}^k}{\arg\min} \frac{1}{n_2} \sum_{i=1}^{n_2} \ell(y_{T+1,i}, \langle \mathbf{w}, \hat{\phi}(\mathbf{x}_{T+1,i}) \rangle).$$

Then, we define the true target risk of the learned linear classifier $\hat{\mathbf{w}}_{T+1}$ as:

$$\mathcal{L}(\hat{\phi}, \hat{\mathbf{w}}_{T+1}) = \mathbb{E}_{(\mathbf{x},y) \sim \mu_{T+1}}[\ell(y, \langle \hat{\mathbf{w}}_{T+1}, \hat{\phi}(\mathbf{x}) \rangle)]$$

and want it to be small and as close as possible to the ideal true risk $\mathcal{L}(\phi^*, \mathbf{w}_{T+1}^*)$ where

$$\forall t \in [[T+1]] \text{ and } (\mathbf{x}, y) \sim \mu_t, \quad y = \langle \mathbf{w}_t^*, \phi^*(\mathbf{x}) \rangle + \varepsilon, \quad \varepsilon \sim \mathcal{N}(0, \sigma^2). \tag{2}$$

Equivalently, most of the works found in the literature seek to upper-bound the *excess risk* defined as $\text{ER}(\hat{\phi}, \hat{\mathbf{w}}_{T+1}) := \mathcal{L}(\hat{\phi}, \hat{\mathbf{w}}_{T+1}) - \mathcal{L}(\phi^*, \mathbf{w}^*_{T+1})$ with quantities involved in the learning process.

**Remark 1** *We note that many popular meta-learning algorithms used for FSL do not follow exactly the approach described above. However, we believe that the exact way of how this is done algorithmically (with or without the support set, with or without learning episodes) does not change the statistical challenge of it which is to learn a model that can provably generalize with little supervision. Supervised learning theory tells us that generalization in this case is poor (not enough target data and it is difficult to rely on data coming from different probability distributions), while the theoretical works we built upon suggest that source data may contribute in improving the generalization of the learned model alongside the target data if the assumptions described below are satisfied.*

## 3    FROM THEORY TO PRACTICE

In this section, we highlight main theoretical contributions that provably ensure the success of meta-learning in improving the performance on the previously unseen target task with the increasing number of source tasks and the amount of data available for them. We then concentrate our attention on the most recent theoretical advances leading to the fastest learning rates and show how the assumptions used to obtain them can be forced in practice through a novel regularization strategy.

### 3.1    WHEN DOES META-LEARNING PROVABLY WORK?

One requirement for meta-learning to succeed in FSC is that a representation learned on meta-train data should be useful for learning a good predictor on the meta-test data set. This is reflected by bounding the excess target risk by a quantity that involves the number of samples in both meta-train and meta-test samples and the number of available meta-train tasks.

To this end, first studies in the context of meta-learning relied on probabilistic assumption (Baxter, 2000; Pentina & Lampert, 2014; Maurer et al., 2016; Amit & Meir, 2018; Yin et al., 2020) stating that meta-train and meta-test tasks distributions are all sampled i.i.d. from the same random distribution. This assumption, however, is considered unrealistic as in FSL source and target tasks' data are often given by different draws (without replacement) from the same dataset. In this setup, the above-mentioned works obtained the bounds having the following form:

$$\text{ER}(\hat{\phi}, \hat{\mathbf{w}}_{T+1}) \leq O\left(\frac{1}{\sqrt{n_1}} + \frac{1}{\sqrt{T}}\right).$$

This guarantee implies that not only the number of source data, but also the number of tasks should be large in order to draw the second term to $0$. An improvement was then proposed by Du et al. (2020) and Tripuraneni et al. (2020) that obtained the bounds on the excess risk behaving as

$$O\left(\frac{kd}{\sqrt{n_1 T}} + \frac{k}{\sqrt{n_2}}\right) \quad \text{and} \quad \tilde{O}\left(\frac{kd}{n_1 T} + \frac{k}{n_2}\right),$$

respectively, where $k \ll d$ is the dimensionality of the learned representation and $\tilde{O}(\cdot)$ hides logarithmic factors. Both these results show that all the source and target samples are useful in minimizing the excess risk: in the FSL regime where target data is scarce, all source data helps to learn well. From a set of assumptions made by the authors in both of these works , we note the following two:

**Assumption 1.** The matrix of optimal predictors $\mathbf{W}^*$ should cover all the directions in $\mathbb{R}^k$ evenly. More formally, this can be stated as

$$R_\sigma(\mathbf{W}^*) = \frac{\sigma_1(\mathbf{W}^*)}{\sigma_k(\mathbf{W}^*)} = O(1), \tag{3}$$

where $\sigma_i(\cdot)$ denotes the $i^{\text{th}}$ singular value of $\mathbf{W}^*$. As pointed out by the authors, such an assumption can be seen as a measure of diversity between the source tasks that are expected to be complementary to each other in order to provide a meaningful representation for a previously unseen target task.

Figure 1: Illustration of the example from Section 3.2 with $\varepsilon = 0.02$.

**Assumption 2.** The norm of the optimal predictors $\mathbf{w}^*$ should not increase with the number of tasks seen during meta-training[2]. This assumption says that the classification margin of linear predictors should remain constant thus avoiding over- or under-specialization to the seen tasks.

While being highly insightful, the authors did not provide any experimental evidence suggesting that verifying these assumptions in practice helps to learn more efficiently in the considered learning setting. To bridge this gap, we propose to use a general regularization scheme that allows us to enforce these assumptions when learning the matrix of predictors in several popular meta-learning algorithms.

### 3.2 PUTTING THEORY TO WORK

As the assumptions mentioned above are stated for the optimal predictors that are inherently linked to the data generating process, one may wonder what happens when these latter do not satisfy them. To this end, we aim to answer the following question:

*Given $\mathbf{W}^*$ such that $R_\sigma(\mathbf{W}^*) \gg 1$, can we learn $\widehat{\mathbf{W}}$ with $R_\sigma(\widehat{\mathbf{W}}) \approx 1$ while solving the underlying classification problems equally well?*

It turns out that we can construct an example illustrated in Fig. 1 for which the answer to this question is positive. To this end, let us consider a binary classification problem over $\mathbb{X} \subseteq \mathbb{R}^3$ with labels $\mathbb{Y} = \{-1, 1\}$ and two source tasks generated for $k, \varepsilon \in ]0, 1]$, as follows:

1. $\mu_1$ is uniform over $\{1 - k\varepsilon, k, 1\} \times \{1\} \cup \{1 + k\varepsilon, k, -1\} \times \{-1\}$;

2. $\mu_2$ is uniform over $\{1 + k\varepsilon, k, \frac{k-1}{\varepsilon}\} \times \{1\} \cup \{-1 + k\varepsilon, k, \frac{1+k}{\varepsilon}\} \times \{-1\}$.

We now define the optimal representation and two optimal predictors for each distribution as the solution to Eq. 1 over the two data generating distributions and $\Phi = \{\phi | \phi(\mathbf{x}) = \mathbf{\Phi}^T \mathbf{x}, \mathbf{\Phi} \in \mathbb{R}^{3 \times 2}\}$:

$$\phi^*, \mathbf{W}^* = \underset{\phi \in \Phi, \mathbf{W} \in \mathbb{R}^{2 \times 2}}{\arg\min} \sum_{i=1}^{2} \underset{(\mathbf{x}, y) \sim \mu_i}{\mathbb{E}} \ell(y, \langle \mathbf{w}_i, \phi(\mathbf{x}) \rangle), \tag{4}$$

One solution to this problem can be given as follows:

$$\mathbf{\Phi}^* = \begin{pmatrix} 1 & 0 & 0 \\ 0 & 1 & 0 \end{pmatrix}^T, \quad \mathbf{W}^* = \begin{pmatrix} 1 & \varepsilon \\ 1 & -\varepsilon \end{pmatrix},$$

where $\phi^*$ projects the data generated by $\mu_i$ to a two-dimensional space by discarding its third dimension and the linear predictors satisfy the data generating process from Eq. 2 with $\varepsilon = 0$. One can verify that in this case $\mathbf{W}^*$ have singular values equal to $\sqrt{2}$ and $\sqrt{2}\varepsilon$, so that the ratio $R_\sigma(\mathbf{W}^*) = \frac{1}{\varepsilon}$: when $\varepsilon \to 0$, the optimal predictors make the ratio arbitrary large thus violating Assumption 1.

---

[2]While not stated as a separate assumption, in Du et al. (2020) assume it to derive the Assumption 1 mentioned above. See p.5 and the discussion after Assumption 4.3 in their pre-print.

Let us now consider a different problem where we want to solve Eq. 4 with a constraint that forces linear predictors to satisfy Assumption 1:

$$\widehat{\phi}, \widehat{\mathbf{W}} = \underset{\phi \in \Phi, \mathbf{W} \in \mathbb{R}^{2 \times 2}}{\arg\min} \sum_{i=1}^{2} \underset{(\mathbf{x}, y) \sim \mu_i}{\mathbb{E}} \ell(y, \langle \mathbf{w}_i, \phi(\mathbf{x}) \rangle), \quad \text{s.t. } R_\sigma(\mathbf{W}) \approx 1. \tag{5}$$

Its solution is different and is given by

$$\widehat{\mathbf{\Phi}} = \begin{pmatrix} 0 & 1 & 0 \\ 0 & 0 & 1 \end{pmatrix}^T, \quad \widehat{\mathbf{W}} = \begin{pmatrix} 0 & 1 \\ 1 & -\varepsilon \end{pmatrix}.$$

Similarly to $\mathbf{\Phi}^*$, $\widehat{\mathbf{\Phi}}$ projects to a two-dimensional space by discarding the first dimension of the data generated by $\mu_i$. The learned predictors in this case also satisfy Eq. 2 with $\varepsilon = 0$, but contrary to $\mathbf{W}^*$, $R_\sigma(\widehat{\mathbf{W}}) = \sqrt{\frac{2 + \varepsilon^2 + \varepsilon\sqrt{\varepsilon^2 + 4}}{2 + \varepsilon^2 - \varepsilon\sqrt{\varepsilon^2 + 4}}}$ tends to 1 when $\varepsilon \to 0$.

Several remarks are in order here. First, it shows that even when $\mathbf{W}^*$ does not satisfy Assumption 1 in the space induced by $\phi^*$, it may still be possible to learn a new representation space $\widehat{\phi}$ such that the optimal predictors in this space will satisfy Assumption 1. This can be done either by considering the constrained problem from Eq. 5, or by using a more common strategy that consists in adding $R_\sigma(\mathbf{W})$ directly as a regularization term

$$\hat{\phi}, \widehat{\mathbf{W}} = \underset{\phi \in \Phi, \mathbf{W} \in \mathbb{R}^{T \times k}}{\arg\min} \frac{1}{2Tn_1} \sum_{t=1}^{T} \sum_{i=1}^{n_1} \ell(y_{t,i}, \langle \mathbf{w}_t, \phi(\mathbf{x}_{t,i}) \rangle) + \lambda_1 R_\sigma(\mathbf{W}). \tag{6}$$

Below, we explain how to implement this idea in practice for popular meta-learning algorithms.

**Ensuring assumption 1.** We propose to compute singular values of $\mathbf{W}$ during the meta-training stage and follow its evolution during the learning episodes. In practice, this can be done by performing the Singular Value Decomposition (SVD) on $\mathbf{W} \in \mathbb{R}^{T \times k}$ with a computational cost of $O(Tk^2)$ floating-point operations (flop). However, as $T$ is typically quite large, we propose a more computationally efficient solution that is to take into account only the last batch of $N$ predictors (with $N \ll T$) grouped in the matrix $\mathbf{W}_N \in \mathbb{R}^{N \times k}$ that capture the latest dynamics in the learning process. We further note that $\sigma_i(\mathbf{W}_N \mathbf{W}_N^\top) = \sigma_i^2(\mathbf{W}_N)$, $\forall i \in [[N]]$ implying that we can calculate the SVD of $\mathbf{W}_N \mathbf{W}_N^\top$ (or $\mathbf{W}_N^\top \mathbf{W}_N$ for $k \leq N$) and retrieve the singular values from it afterwards.

We now want to verify whether the optimal linear predictors $\mathbf{w}_t$ cover all directions in the embedding space by tracking the evolution of the ratio of singular values $R_\sigma(\mathbf{W}_N)$ during the training process. For the sake of conciseness, we use $R_\sigma$ instead of $R_\sigma(\mathbf{W}_N)$ thereafter. According to the theory, we expect $R_\sigma$ to decrease during training thus improving the generalization of the learned predictors and preparing them for the target task. When we want to enforce such a behavior in practice, we propose to use $R_\sigma$ as a regularization term in the training loss of popular meta-learning algorithms.

Alternatively, as the smallest singular value $\sigma_N(\mathbf{W}_N)$ can be close to 0 and lead to numerical errors, we propose to replace the ratio of the vector of singular values by its entropy as follows:

$$H_\sigma(\mathbf{W}_N) = - \sum_{i=1}^{N} \text{softmax}(\sigma(\mathbf{W}_N))_i \cdot \log \text{softmax}(\sigma(\mathbf{W}_N))_i,$$

where $\text{softmax}(\cdot)_i$ is the $i^{th}$ output of the softmax function. As with $R_\sigma$, we write $H_\sigma$ instead of $H_\sigma(\mathbf{W}_N)$ from now on. Since uniform distribution has the highest entropy, regularizing with $R_\sigma$ or $-H_\sigma$ leads to a better coverage of $\mathbb{R}^k$ by ensuring a nearly identical importance regardless of the direction. We refer the reader to the Supplementary materials for the derivations ensuring the existence of the subgradients for these terms.

**Ensuring assumption 2.** In addition to the full coverage of the embedding space by the linear predictors, the meta-learning theory assumes that the norm of the linear predictors does not increase with the number of tasks seen during meta-training, *i.e.*, $\|\mathbf{w}\|_2 = O(1)$ or, equivalently, $\|\mathbf{W}\|_F^2 = O(T)$. If this assumption does not hold in practice, we propose to regularize the norm of linear predictors during training or directly normalize the obtained linear predictors $\bar{\mathbf{w}} = \frac{\mathbf{w}}{\|\mathbf{w}\|_2}$.

The final meta-training loss with the theory-inspired regularization terms is given as:

$$\min_{\phi\in\Phi,\mathbf{W}\in\mathbb{R}^{T\times k}} \frac{1}{2Tn_1}\sum_{t=1}^{T}\sum_{i=1}^{n_1}\ell(y_{t,i},\langle\mathbf{w}_t,\phi(\mathbf{x}_{t,i})\rangle) + \lambda_1 R_\sigma(\mathbf{W}_N) + \lambda_2\|\mathbf{W}_N\|_F^2, \qquad (7)$$

and depending on the considered algorithm, we can replace $R_\sigma$ by $-H_\sigma$ and/or replace $\mathbf{w}_t$ by $\bar{\mathbf{w}}_t$ instead of regularizing with $\|\mathbf{W}_N\|_F^2$. In what follows, we consider $\lambda_1 = \lambda_2 = 1$ and we refer the reader to the Supplementary materials for more details and experiments with other values.

To the best of our knowledge, such regularization terms based on insights from the advances in meta-learning theory have never been used in the literature before. We also further use the basic quantities involved in the proposed regularization terms as indicators of whether a given meta-learning algorithm naturally satisfies the assumptions ensuring an efficient meta-learning in practice or not.

### 3.3 RELATED WORK

Below, we discuss several related studies aiming at improving the general understanding of meta-learning, and mention other regularization terms specifically designed for meta-learning.

**Understanding meta-learning** While a complete theory for meta-learning is still lacking, several recent works aimed to shed light on phenomena commonly observed in meta-learning by evaluating different intuitive heuristics. For instance, Raghu et al. (2020) investigated whether the popular gradient-based MAML algorithm relies on rapid learning with significant changes in the representations when deployed on target task, or due to feature reuse where the learned representation remains almost intact. They establish that the latter factor is dominant and propose a new variation of MAML that freezes all but task-specific layers of the neural network when learning new tasks. In another study (Goldblum et al., 2020) the authors explain the success of meta-learning approaches by their capability to either cluster classes more tightly in feature space (task-specific adaptation approach), or to search for meta-parameters that lie close in weight space to many task-specific minima (full fine-tuning approach). Finally, the effect of the number of shots on the classification accuracy was studied theoretically and illustrated empirically in Cao et al. (2020) for the popular metric-based PROTONET algorithm. Our paper is complementary to all other works mentioned above as it investigates a new aspect of meta-learning that has never been studied before, while following a sound theory. Also, we provide a more complete experimental evaluation as the three different approaches of meta-learning (based on gradient, metric or transfer learning), separately presented in Raghu et al. (2020), Cao et al. (2020) and Goldblum et al. (2020), are now compared together.

**Other regularization strategies** Regularization is a common tool to reduce model complexity during learning for better generalization, and the variations of its two most famous instances given by weight decay (Krogh & Hertz, 1992) and dropout (Srivastava et al., 2014) are commonly used as a basis in meta-learning literature as well. In general, regularization in meta-learning is applied to the weights of the whole neural network (Balaji et al., 2018; Yin et al., 2020), the predictions (Jamal & Qi, 2019; Goldblum et al., 2020) or is introduced via a prior hypothesis biased regularized empirical risk minimization (Pentina & Lampert, 2014; Kuzborskij & Orabona, 2017; Denevi et al., 2018a;b; 2019). Our proposal is different from all the approaches mentioned above for the following reasons. First, we do not regularize the whole weight matrix learned by the neural network but the linear predictors of its last layer contrary to what was done in the methods of the first group, and, more specifically, the famous weight decay approach (Krogh & Hertz, 1992). The purpose of the regularization in our case is also completely different: weight decay is used to improve generalization through sparsity in order to avoid overfitting, while our goal is to keep the classification margin unchanged during the training to avoid over-/under-specialization to some source tasks. Similarly, spectral normalization proposed by Miyato et al. (2018) to satisfy the Lipschitz constraint in GANs through dividing $\mathbf{W}$ values by $\sigma_{\max}(\mathbf{W})$ does not affect the ratio between $\sigma_{\max}(\mathbf{W})$ and $\sigma_{\min}(\mathbf{W})$ and serves a completely different purpose. Second, we regularize the singular values (entropy or ratio) of the matrix of linear predictors instead of the predictions, as done by the methods of the second group (e.g., using the theoretic-information quantities in Jamal & Qi (2019) and Yin et al. (2020)). Finally, the works of the last group are related to the online setting with convex loss functions only, and, similarly to the algorithms from the second group, do not specifically target the spectral properties of the learned predictors. Last, but not least, our proposal is built upon the most recent advances in the meta-learning field leading to faster learning rates contrary to previous works.

## 4 PRACTICAL RESULTS

In this section, we use extensive experimental evaluations to answer the following two questions:

**Q1)** Do popular meta-learning methods naturally satisfy the learning bounds assumptions?

**Q2)** Does ensuring these assumptions help to (meta-)learn more efficiently?

For **Q1**, we run the original implementations of popular meta-learning methods to see what is their natural behavior. For **Q2**, we study the impact of forcing them to closely follow the theoretical setup.

### 4.1 EXPERIMENTAL SETUP

**Datasets & Baselines** We consider few-shot image classification problem on three benchmark datasets, namely: 1) Omniglot (Lake et al., 2015) consisting of $1{,}623$ classes with 20 images/class of size $28 \times 28$; 2) miniImageNet (Ravi & Larochelle, 2017) consisting of $100$ classes with $600$ images of size $84 \times 84$ per class and 3) tieredImageNet (Ren et al., 2018) consisting of $779{,}165$ images divided into $608$ classes. For each dataset, we follow the commonly adopted experimental protocol used in Finn et al. (2017) and Chen et al. (2019) and use a four-layer convolution backbone (Conv-4) with 64 filters as done by Chen et al. (2019). On Omniglot, we perform 20-way classification with 1 shot and 5 shots, while on miniImageNet and tieredImageNet we perform 5-way classification with 1 shot and 5 shots. Finally, we evaluate four FSL methods: two popular meta-learning strategies, namely, MAML (Finn et al., 2017), a gradient-based method, and Prototypical Networks (PROTONET) (Snell et al., 2017), a metric-based approach; two popular transfer learning baselines, termed as BASELINE and BASELINE++ (Ravi & Larochelle, 2017; Gidaris & Komodakis, 2018; Chen et al., 2019). Even though these baselines are trained with the standard supervised learning framework, such a training can also be seen as learning a single task in the LTL framework.

**Implementation details** Enforcing Assumptions 1 and 2 for MAML is straightforward as it closely follows the LTL framework of episodic training. For each task, the model learns a batch of linear predictors and we can directly take them as $\mathbf{W}_N$ to compute its SVD. Since the linear predictors are the weights of our model and change slowly, regularizing the norm $\|\mathbf{W}_N\|_F$ and the ratio of singular values $R_\sigma$ does not cause instabilities during training. Meanwhile, metric-based methods do not use linear predictors but compute a similarity between features. In the case of PROTONET, the similarity is computed with respect to class prototypes (*i.e.* the mean features of the images of each class). Since they act as linear predictors, a first idea would be to regularize the norm and ratio of singular values of the prototypes. Unfortunately, this latter strategy hinders the convergence of the network and leads to numerical instabilities. Most likely because prototypes are computed from image features which suffer from rapid changes across batches. Consequently, we regularize the entropy of singular values $H_\sigma$ instead of the ratio $R_\sigma$ to avoid instabilities during training to ensure Assumption 1 and we normalize the prototypes to ensure Assumption 2 by replacing $\mathbf{w}_t$ with $\bar{\mathbf{w}}_t$ in Eq. 7. For transfer learning methods BASELINE and BASELINE++, the last layer of the network is discarded and linear predictors are learned during meta-testing. Thus, we only regularize the norm $\|\mathbf{W}_N\|_F$ of predictors learned during the finetuning phase of meta-testing. Similarly to MAML, we compute $R_\sigma$ with the last layer of the network during training and fine-tuning phase.

**Remark 2** *We choose well-established meta-learning algorithms for our comparison, but the proposed regularization can be integrated similarly into their recent variations (Park & Oliva, 2019; Lee et al., 2019) (see Supplementary materials for results obtained with the method of Park & Oliva (2019)). Finally, using models that do not rely on linear predictors is also possible but might be more difficult as it would require upstream work to understand which part of the model acts as predictors (as done for PROTONET in this paper) and how to compute and track the desired quantities.*

### 4.2 INSIGHTS

**Q1 – Verifying the assumptions** According to theory, $\|\mathbf{W}_N\|_F$ and $R_\sigma$ should remain constant or converge toward a constant value when monitoring the last $N$ tasks. From Fig. 2(a), we can see that for MAML (Fig. 2(a) top), both $\|\mathbf{W}_N\|_F$ and $R_\sigma$ increase with the number of tasks seen during training, whereas PROTONET (Fig. 2(a) bottom) naturally learns the prototypes with a good coverage of the embedding space, and minimizes their norm. This behavior is rather peculiar as neither

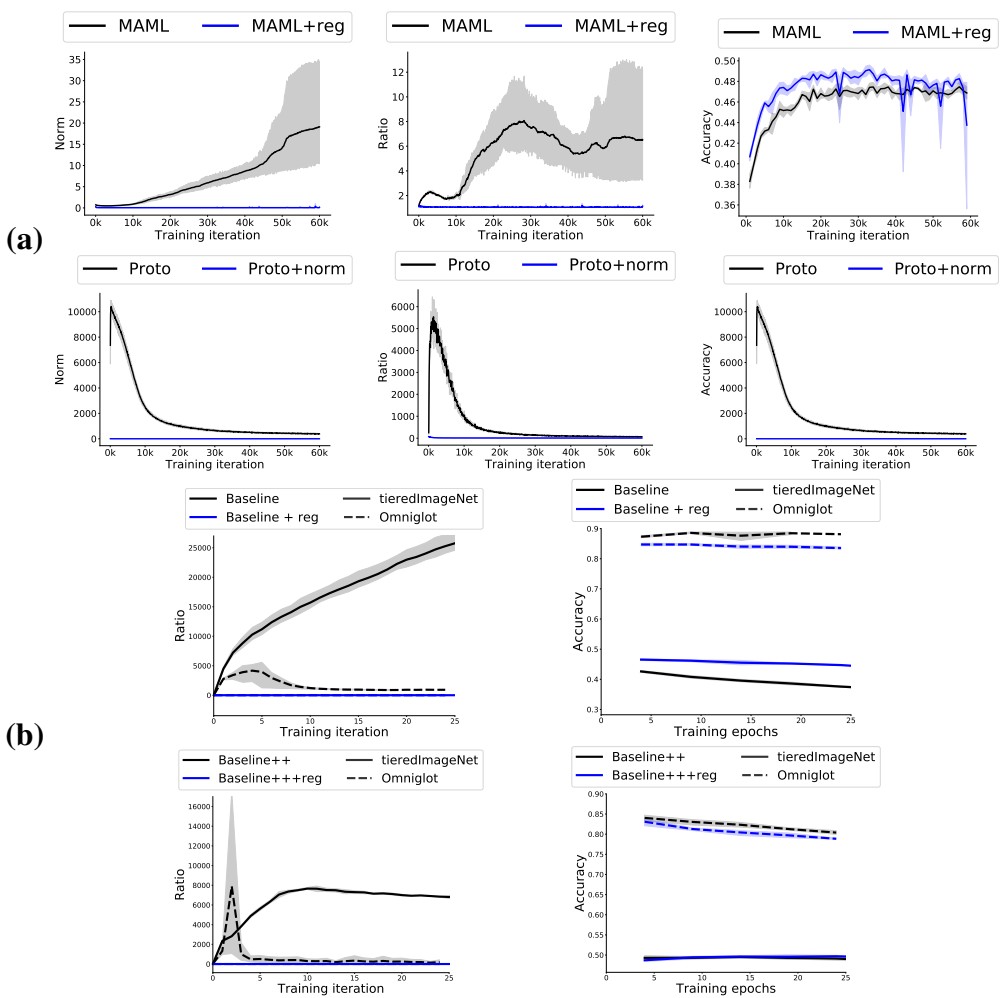

Figure 2: **(a)** Evolution of $\|\mathbf{W}_N\|_F$ (*left*), $R_\sigma$ (*middle*) and validation accuracy (*right*) when training of MAML (*top*) and PROTONET (*bottom*) on miniImageNet (1 shot for MAML, 5 shots for PROTONET). **(b)** Evolution of $R_\sigma$ (*left*) and validation accuracy (*right*) when training BASELINE (*top*) and BASELINE++ (*bottom*) on Omniglot (*dashed lines*) and tieredImageNet (*solid lines*). All training curves were averaged over 4 different random seeds. For MAML, $\|\mathbf{W}_N\|_F$ and $R_\sigma$ increase during training and violate Assumptions 1-2. PROTONET prototypes naturally cover the embedding space, while minimizing their norms. $R_\sigma$ converges during training on both datasets for BASE-LINE++ (similarly to PROTONET) whereas it diverges for BASELINE on tieredImageNet. With our regularization, $\|\mathbf{W}_N\|_F$ and $R_\sigma$ are constant during training in accordance with theory.

of the two methods specifically controls the theoretical quantities of interest, and still, PROTONET manages to do it implicitly. As for the transfer learning baselines (Fig. 2(b) top and bottom), we expect them to learn features that cover the embedding space with $R_\sigma$ rapidly converging towards a constant value. As can be seen in Fig. 2(b), similarly to PROTONET, BASELINE++ naturally learns linear predictors that cover the embedding space. As for BASELINE, it learns a good coverage for Omniglot dataset, but fails to do so for the more complicated tieredImageNet dataset. The observed behavior of these different methods leads to a conclusion that some meta-learning algorithms are inherently more explorative of the embedding space.

**Q2 – Ensuring the assumptions** Armed with our regularization terms, we now aim to force the considered algorithms to verify the assumptions when it is not naturally done. In particular, for MAML we regularize both $\|\mathbf{W}_N\|_F$ and $R_\sigma$ in order to keep them constant throughout the training. Similarly, we regularize $R_\sigma$ during the training of BASELINE and BASELINE++, and both $\|\mathbf{W}_N\|_F$ and $R_\sigma$ during the finetuning phase of meta-testing. For PROTONET, we enforce a normalization of

| Dataset | Episodes | MAML | PROTONET | BASELINE | BASELINE++ |
|---|---|---|---|---|---|
| Omniglot | 20-way 1-shot | +3.95* | +0.33* | −13.2* | −7.29* |
| | 20-way 5-shot | +1.17* | +0.01 | +0.66* | −2.24* |
| miniImageNet | 5-way 1-shot | +1.23* | +0.76* | +1.52* | +0.39 |
| | 5-way 5-shot | +1.96* | +2.03* | +1.66* | −0.13 |
| tieredImageNet | 5-way 1-shot | +1.42* | +2.10* | +5.43* | +0.28 |
| | 5-way 5-shot | +2.66* | +0.23 | +1.92* | −0.72 |

Table 1: Accuracy gap (in p.p.) of the considered algorithms when using the regularization (or normalization in the case of PROTONET) enforcing the theoretical assumptions. All accuracy results are averaged over 2400 test episodes and 4 different seeds. Statistically significant results (out of confidence intervals) are reported with *. Exact performances are on par with those found in the literature and are reported in the Supplementary materials.

the prototypes. According to our results for Q1, regularizing the singular values of the prototypes through the entropy $H_\sigma$ is not necessary.[3] Based on the obtained results, we can make the following conclusions. First, from Fig. 2(a) (left, middle) and Fig. 2(b) (left), we note that for all methods considered, our proposed methodology used to enforce the theoretical assumptions works as expected, and leads to a desired behavior during the learning process. This means that the differences in terms of results presented in Table 1 are explained fully by this particular addition to the optimized objective function. Second, from the shape of the accuracy curves provided in Fig. 2(a) (right) and the accuracy gaps when enforcing the assumptions given in Table 1, we can see that respecting the assumptions leads to several significant improvements related to different aspects of learning. On the one hand, we observe that the final validation accuracy improves significantly in all benchmarks for meta-learning methods and in most of experiments for BASELINE (except for Omniglot, where BASELINE already learns to regularize its linear predictors). In accordance with the theory, we attribute the improvements to the fact that we fully utilize the training data which leads to a tighter bound on the excess target risk and, consequently, to a better generalization performance. On the other hand, we also note that our regularization reduces the sample complexity of learning the target task, as indicated by the faster increase of the validation accuracy from the very beginning of the meta-training. Roughly speaking, less meta-training data is necessary to achieve a performance comparable to that obtained without the proposed regularization using more tasks. Finally, we note that BASELINE++ and PROTONET methods naturally satisfy some assumptions: both learn diverse linear predictors by design, while BASELINE++ also normalizes the weights of its linear predictors. Thus, these methods do not benefit from additional regularization as explained before.

## 5 CONCLUSION

In this paper, we studied the validity of the theoretical assumptions made in recent papers applied to popular meta-learning algorithms and proposed practical ways of enforcing them.

On the one hand, we showed that depending on the problem and algorithm, some models can naturally fulfill the theoretical conditions during training. Some algorithms offer a better covering of the embedding space than others. On the other hand, when the conditions are not verified, learning with our proposed regularization terms allows to learn faster and improve the generalization capabilities of meta-learning methods. The theoretical framework studied in this paper explains the observed performance gain. Notice that no specific hyperparameter tuning was performed as we rather aim at showing the effect of ensuring learning bounds assumptions than comparing performance of the methods. Absolute accuracy results are detailed in the Supplementary materials.

While this paper proposes an initial approach to bridging the gap between theory and practice in meta-learning, some questions remain open on the inner workings of these algorithms. In particular, being able to take better advantage of the particularities of the training tasks during meta-training could help improve the effectiveness of these approaches. Self-supervised meta-learning and multiple target tasks prediction are also important future perspectives for the application of meta-learning.

---

[3]The effect of entropic regularization on PROTONET is detailed in the Supplementary materials.

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
