# OpenReview forum: "Putting Theory to Work: From Learning Bounds to Meta-Learning Algorithms"
_ICLR.cc/2021/Conference — Reject_

### Official Review · AnonReviewer1 · 2020-10-26
**Putting Theory to Work: From Learning Bounds to Meta-Learning Algorithms**

**Rating:** 5
**Confidence:** 4

**Review:**

Summary:
In this paper, the authors aim at bridging the gap between the practice and theory in meta-learning approaches. Specifically, they propose two regularization terms to 1) capture the diversity of the tasks and 2) control the norm of the prediction layer, thereby satisfying the assumptions in meta-learning theory.

Strength:
+ The motivation of this paper is interesting, before proposing the methodology. These theoretical assumptions have not been paid enough attention before.
+ The paper is well-organized and clearly written.
+ The experimental setting is designed in a good manner and the results are promising.

Weakness:
- I am skeptical of the novelty of the second regularize in Eq.(4). According to Section 3.2, it is equivalent to ||w||_{2}=O(1). So what is its difference to a simple l2 weight decay?
- According to Section 2, the outer-level parameters are restricted as a linear layer. Is this means the proposed regularizes would become trivial while applied on top of a more complicated model, e.g., LEO[1]?
- Too few competitors. It would be better to add some comparisons with recent methods.
- The details to calculate the subgradients of the singular values, which is quite complicated, are missing. Especially seeing that there is no guarantee that an auto-differentiation tool will do that correct.

Ref:
[1] Andrei A. Rusu, Dushyant Rao, Jakub Sygnowski, Oriol Vinyals, Razvan Pascanu, Simon Osindero, Raia Hadsell: Meta-Learning with Latent Embedding Optimization. ICLR 2019

Above all, since the contribution and the technical details to calculate the subgradients are not clear to me, I have to currently recommend a weak reject.

---

> ### Author Response · Authors · 2020-11-14
> **Our goal is not to propose a new regularization that outperforms the state of the art**
>
> We thank the reviewer for the detailed and helpful review. We want to make it clear that our goal is not to propose novel meta-learning algorithm with a new regularization that outperforms the state of the art meta-learning methods but rather to find out whether recent theoretical insights from few-shot learning theory are useful in practice. We will adjust the narrative of our paper accordingly to reflect this.
>
> - As explained in Section 3.3 of our paper, normalizing the norm of the linear predictors is different from weight decay as we only regularize/normalize the norm of the linear predictors and not the weights of the whole model. Also, the overall purpose of this in our case is completely different: weight decay is used to improve generalization though sparsity in order to avoid overfitting, while our goal is to keep the classification margin unchanged through the learning process to avoid over/under specialization to some source tasks seen during training. Finally, as the few-shot learning theory suggests, satysfying *both* assumptions is crucial as and only one of them is not enough to ensure efficient few-shot learning. This agrees with the experimental results provided in Table 5 of the Supplementary materials highlighting this latter finding. We added this explanation in Section 3.3.
>
> - While the works on few-shot learning theory consider linear predictors, we agree with the reviewer that in practice the used predictors can be much more complicated and/or different than a linear layer. However, we do not fully understand why the proposed regularization might become trivial with a more complicated model and would appreciate more comments on this. Verifying the assumptions for more complicated models might be more difficult because it would require upstream work to understand which part of the model acts as predictors (we have already done it for ProtoNet that does not use a linear layer for classification) and how to compute and track the desired quantities. We added this explanation in Remark 2.
>
> - We agree that it may be interesting to add more recent and complicated few-shot learning methods to our comparison and we are working on it for the revised version of the manuscript and we will provide additional comparisons as soon as possible. We note, however, that considering established efficient methods appears to be more appropriate as most of the more complicated methods follow similar methodology (e.g. Meta-Curvature [1], MetaOptNet [2]). We added this explanation in Remark 2.
>
> - Computing the SVD is entirely differentiable and naturally supported in auto-differentiation frameworks such as Pytorch and Tensorflow and backpropagation through SVD was already used in [3].
>
> [1]: E. Park, J. Oliva. Meta-Curvature, NeurIPS 2019
> [2]: K. Lee, S. Maji, A. Ravichandran, S. Soatto. Meta-Learning with Differentiable Convex Optimization, CVPR 2019
> [3]: X. Chen, S. Wang, B. Fu, M. Long, J. Wang. Catastrophic Forgetting Meets Negative Transfer:Batch Spectral Shrinkage for Safe Transfer Learning, NeurIPS 2019

---

### Official Review · AnonReviewer4 · 2020-10-26
**Improving practical performance of meta-learning, with inspiration from theoretical results**

**Rating:** 5
**Confidence:** 3

**Review:**

To improve the practical performance of meta-learning algorithms, this paper proposes two regularization terms that are motivated by two common assumptions in some recent theoretical work on meta-learning, namely (1) the optimal (linear) predictors cover the embedding space evenly, and (2) the norms of the optimal predictors remain bounded as the number of tasks grow. Numerical experiments show that the proposed regularization terms help achieve better performance of meta-learning in some tasks.

This work serves as a nice attempt to instruct the practice of meta-learning with theoretical insights. Below are some of my concerns.

- In some experimental results, the improvement due to the proposed regularization seems to be at the same level of the standard deviation, as well as the difference between the reproduced results of existing meta-learning algorithms and those reported in earlier papers. This casts doubt on the true efficacy of the proposed methods.

- For the loss function in Eq. (4), it is more reasonable and natural to introduce two weighting parameters (as tunable hyperparameters) for the proposed regularization terms.

- The authors often talk about "enforcing/ensuring the assumptions". However, from my understanding, whether the assumptions (on the optimal linear predictors, or "ground-truth" predictors) hold or not depends on the learning problem itself, NOT on the algorithms. Therefore, there is no way we can enforce/ensure these assumptions. I would prefer using the phrase "respecting the assumptions" (used by the authors on Page 8); this seems more accurate and reasonable.

- Following the previous point, I'm curious about one question: if the learning problem actually doesn't satisfy the two assumptions, then is it still helpful to add the proposed regularization terms to the loss function? (I'm not sure, but my guess is no; indeed, it might even hurt.) To solve puzzles like this, I would encourage the authors to conduct some synthetic experiments, where they can design the data generating process (e.g. they can control whether the true linear predictors satisfy the assumptions or not). Since this work is a connection between theory and practice, I believe that experiments with synthetic data can help explain things more clearly and make the claims more convincing.

---

> ### Author Response · Authors · 2020-11-14
> **Our goal is not to propose a new regularization that outperforms the state of the art**
>
> We thank the reviewer for the review.
>
> - Learning with deep neural networks that optimize a non-convex objective function often leads to differences between the reported and the reproduced results even when using the code provided by authors. Thus, it is a common practice to compare the obtained results with the reproduced results rather than the reported ones [1,2,3]. Beyond that, we are interested in the relative difference  between our reproduced results and those obtained with regularization that follows the current theory of few-shot learning. These differences are observed consistently in the experiments repeated 4 times with 4 different seeds and they are  *statiscally significant* when marked with "\*" in Table 1, which means that the results are outside of the standard deviation observed.
>
> - Indeed, for better results, it is natural to introduce hyperparameters to weight the regularization terms. However, our goal is not to propose novel meta-learning algorithm with a new regularization that outperforms the state of the art meta-learning methods but rather to find out whether recent theoretical insights from few-shot learning theory are useful in practice. We will make sure to adjust the narrative accordingly and we will add additional experiments highlighting the results obtained with hyperparameter tuning.
>
> - As many other theoretical results in the statistical learning literature, the assumption given in Eq. 3 is stated for the true optimal matrix of the linear predictors $W^*$ which is unknown in practice. However, one can assume that the meta-learning process leads to a consistent estimation of $W^*$ and expect the output matrix $\widehat{W}$ to be close to the latter and thus, to satisfy the same assumptions too. We also agree with the reviewer regarding the employed terminology as our primary goal was indeed to verify whether the theoretical assumptions hold and to find practical ways to "ensure" them when it is not the case. We added this explanation in Section 3.2.
>
> - The question asked by the reviewer is very interesting as indeed, if the optimal predictors are not diverse enough, then we should not expect that the source data will be helpful in reducing the excess risk on the previously unseen target task. In practice, however, we deal with empirical estimators that, contrary to the theoretical setup, may be forced to lie outside the true unregularized argmin set through our regularization. We hypothesize that it may be possible to sacrifice some accuracy by learning less efficiently on the source tasks to have a better performance on the target task. We also agree that it would be interesting to find an illustrative synthetic experiment for this and we are currently working on providing it in addition to new illustrative figures that we have already included in the revised version of the manuscript in Figure 1.
>
> [1]: HOW TO TRAIN YOUR MAML, ICLR 2019
> [2]: A CLOSER LOOK AT FEW-SHOT CLASSIFICATION, ICLR 2019
> [3]: RAPID LEARNING OR FEATURE REUSE? TOWARDS UNDERSTANDING THE EFFECTIVENESS OF MAML, ICLR 2020

---

> > ### Author Response · Authors · 2020-11-24
> > **Latest revision**
> >
> > As a follow-up on our previous comments:
> > - We adjusted the objective function to introduce hyperparameters to weight the regularization terms and we added additional experiments highlighting the results obtained when tuning them in the Supplementary Materials (Table 9 and Table 10).
> > - We thank the reviewer for its last question as it allowed us to come up with an example that answers it and better justifies our contribution. In short, the solution to the original problem with $W^*$ and $\phi^*$ may be forced to lie outside the unregularized argmin set through our regularization. We provide an example for it in the beginning of Section 3.2 and illustrate it in Figure 1.

---

### Official Review · AnonReviewer3 · 2020-10-28
**A theory inspired method for meta-learning**

**Rating:** 4
**Confidence:** 4

**Review:**

The main motivation of this paper is based on the theoretical results of meta-learning. To ensure the assumptions of the theories, the authors propose a novel regularizer, which improves the generalization ability of the model. Some results on few-shot learning benchmarks show the proposed method improves w.r.t. those baselines.

Here are the main concerns of this paper:
1. The proposed method in this paper is based on the meta-learning theory as stated in Section 2. However, the theoretical setting here is not fully consistent with the few-shot learning setting. For example, there is no validation set in Eq. 1. The authors should make more discussions here to show will these differences influence the final results.
2. One main theoretical assumption in meta-learning theory is the task distribution. Could the authors make this notion clear? Should we do empirical results on those tasks with different kinds of task distributions?
3. The meta-learning loss in Eq. 4 is a bit different from the popular meta-learning objective. For example, in MAML, we do not optimize the classifier W till convergence while only a limited number of gradient steps are used.
4. The authors should list those baseline values in Table 1, which are still important for reference.

---

> ### Author Response · Authors · 2020-11-14
> **Our goal is to study the theoretical vs real behavior of meta-learning algorithms**
>
> We thank the reviewer for the review. We want to make it clear that our goal is not to propose novel meta-learning algorithm with a new regularization that outperforms the state of the art meta-learning methods but rather to find out whether recent theoretical insights from few-shot learning theory are useful in practice and coherent with the real-world behavior of several popular meta-learning algorithms. We will adjust the narrative of our paper accordingly.
>
> 1. In the context of our work, one should understand few-shot learning as a theoretical setup considered in Du et al.'19 where we are given a set of source tasks, and we want to make the most of them to learn efficiently (in the sample complexity sense) a new target task with few labeled data. Note that the exact way of how this is done algorithmically (with or without the support set, with or without learning episodes) does not change the statistical learning challenge of it which is to learn a model that can generalize with little supervision. Traditional statistical learning theory tells us that the generalization in this case will be provably poor (not enough target data and impossible to rely on data coming from different probability distributions), while the theoretical works we built upon tell us that source data may contribute equally in improving the generalization of the learned model alongside the target data if the assumptions that we study are respected.
> Apart from that, we agree with the reviewer on another important point: few-shot learning (FSL) is not strictly equivalent to meta-learning, even though the latter is almost always evaluated on the former task. We added this explanation in Remark 1.
>
> 2. Indeed, the assumption regarding the task distribution is crucial in the previous works on meta-learning and few-shot classification. One should think of the i.i.d assumption used in Maurer et al.'16 in the same sense as if it were related to the random vectors and not probability distributions: if it holds, then the distributions of all source and target tasks are independent and follow the same random distribution. This assumption is not realistic in practice as the source tasks in few-shot classification are often dependent as they usually belong to different draws (without replacement) from the same dataset. We added this explanation in Section 3.1.
>
> 3. We agree with the reviewer on the fact that the theoretical setups of Maurer et al.'16 and Du et al.'20 do not exactly correspond to the MAML algorithm. As with any theory, its application in practice requires certain relaxations of the considered setup which correspond in our case to assuming that the algorithmic details of how the learning in a few-shot regime is achieved should not impact the general conditions that should be respected in order for it to succeed. We added this explanation in Remark 1.
>
> 4. We ask the reviewer to kindly specify to which "those" values he/she is referring to in the last item in the review and we will include them (or point out to where one can find them in the supplementary material) consequently.

---

### Official Review · AnonReviewer2 · 2020-10-28
**The idea of bridging theory and practice is good, but the proposed regularization is not novel.**

**Rating:** 4
**Confidence:** 4

**Review:**

##########################################################################
Summary:

The paper reviews common assumptions made by recent theoretical analysis of meta-learning and applies them to meta-learning methods as regularization. Results show that these regularization terms improve over vanilla meta-learning.

##########################################################################
Reasons for score:

Overall, I vote for reject. The main idea of applying theory to practice is reasonable, but the regularization methods proposed are mainly known. Regularizing the singular value is similar to the spectral normalization proposed in [1]. The Frobenius norm regularization is similar to the commonly used weight decay.

##########################################################################
1.	Assumption 1 in Du et al. states that the ground truth weight should cover all directions evenly. It cannot be ensured when the tasks are fixed. The proposed regularization penalizes the condition number of the weight matrix during training, which is more similar to the spectral normalization proposed in [1]. As to regularizing the Frobenius norm, there exist a line of literature showing weight decay works for general settings apart from meta-learning. Thus, I think the regularization proposed in this paper is known.
2.	The experimental results indeed improve over vanilla meta-learning. However, as shown in [2], even by with some simple tricks, meta-learning can be more stable and achieves better results. This casts doubt on the value of the proposed method.

[1] Spectral Normalization for Generative Adversarial Networks, ICLR 2018
[2] HOW TO TRAIN YOUR MAML, ICLR 2019

---

> ### Author Response · Authors · 2020-11-14
> **Our regularization is fundamentally different, and the goal is to study the theoretical vs real behavior of meta-learning algorithms**
>
> We thank the reviewer for the feedback. Before addressing the different concerns raised by the reviewer, we first want to insist that our goal is not to propose a novel regularization that outperforms the state of the art few-shot classification methods but rather to study whether current theoretical results leading to provably efficient few-shot classification agree with the real-world behaviour of several popular meta-learning algorithms. Note that we do not seek to improve the performance or to show that our regularization works better than other methods: it is used solely as a way of verifying whether theoretical assumptions are useful, to some extent, in practice. We will make sure to adjust the wording accordingly.
>
> 1. a.  As many other theoretical results in the statistical learning literature, the assumption given in Eq. 3 is stated for the true optimal matrix of the linear predictors $W^*$ which is unknown in practice. However, one can assume that the meta-learning process leads to a consistent estimation of $W^*$ and expect the output matrix $\widehat{W}$ to be close to the latter and thus, to satisfy the same assumptions too. We added this explanation in Section 3.2.
>
>  b.  Our regularization is strictly different from [1] from the algebraic point of view as dividing the $\widehat{W}$ values by $\sigma_{max}$ as done in [1] does not affect the ratio between $\sigma_{max}$ and $\sigma_{min}$. This trivially follows from the fact that if $\sigma_{min} \neq \sigma_{max}$ then $\widehat{W} / \sigma_{max} = Udiag(\{1, \dots, \sigma_{min}/\sigma_{max}\})V$ and $1 \neq \sigma_{min}/\sigma_{max}$ ! Also, the regularization in [1] is used in GANs to satisfy the Lipschitz constraint which has nothing to do with our goal of increasing the diversity of linear predictors. We added this explanation in Section 3.3.
>
>  c.  As explained in Section 3.3 of our paper, normalizing the norm of the linear predictors is different from weight decay as we only regularize/normalize the norm of the linear predictors and not the weights of the whole model. Also, the overall purpose of this in our case is completely different: weight decay is used to improve generalization though sparsity in order to avoid overfitting, while our goal is to keep the classification margin unchanged through the learning process to avoid over/under specialization to some source tasks seen during the training. We added this explanation in Section 3.3.
>
> 2.  MAML++ [2] improves over vanilla MAML using *implementation* tricks such as learning rate annealing and per-step batch normalization. This has no strict theoretical justification and bears no connection to our proposal. We choose to first verify the theoretical insights on the most established methods from the field before embarking on adding the most recent contributions.
>
> [1] Spectral Normalization for Generative Adversarial Networks, ICLR 2018
> [2] HOW TO TRAIN YOUR MAML, ICLR 2019

---

### Author Response · Authors · 2020-11-14
**Clarification of several common remarks**

We thank the reviewers for their comments. Before answering each reviewer individually, we would like to clarify several common remarks made by the reviewers.

1. "The proposed regularization is not novel and is similar to weight decay and spectral normalization". We would like to insist on the fact that these two regularizations are *fundamentally* different from ours. For the former, we note that weight decay regularizes the whole weight matrix learned by the neural network to improve generalization and avoid overfitting though sparsity, while our goal is to keep the classification margin unchanged during the training to avoid over-/under-specialization to some source tasks. Similarly, spectral normalization proposed by Miyato et al. ICLR 2018 to satisfy the Lipschitz constraint in GANs through dividing $W^*$ values by $\sigma_\text{max}(W^*)$ serves a completely different purpose and does not impact the considered ratio as explained in the revised version of our manuscript (see Section 3.3).

2. "The improvement is not significant and more competitors should be considered". We investigate whether few-shot learning theory is supported by empirical observations. We do not seek to improve the classification accuracy through regularization (we do not even tune hyper-parameters!): this is merely a by-product of showing that few-shot learning theory indeed seems to work in practice! The difference in terms of performances are *statistically significant* in all cases when there is a perceivable difference (not necessarily improvement) in terms of the obtained results. As for the number of baselines, we combined several different approaches to few-shot classification studied separately in Cao et al. ICLR'20 (ProtoNet only, same benchmarks), Raghu et al. ICLR'20 (MAML only, miniImageNet + Omniglot) thus providing a more extensive evaluation compared to previous works studying the inner workings of meta-learning published at last year's ICLR.

3. "The assumptions are based on the optimal predictors and thus cannot be ensured." As many other theoretical results in the statistical learning literature, the assumption given in Eq. 3 is stated for the true optimal matrix of the linear predictors $W^*$ which is unknown in practice. However, one can assume that the meta-learning process leads to a consistent estimation of $W^*$ and expect the output matrix $\hat{W}$ to be close to the latter and thus, to satisfy the same assumptions too. We added this explanation in Section 3.2.

---

> ### Author Response · Authors · 2020-11-24
> **Latest Revision: Synthetic example and more recent baseline**
>
> 1. "The proposed regularization is not novel and is similar to weight decay and spectral normalization." It is important to think about the regularization terms as a whole, and to not take the terms separately because satisfying both assumptions is crucial and only one of them is not enough to ensure efficient few-shot learning. In Table 5 of the Supplementary Materials, we showed that applying only the $L_2$ penalty on the linear predictors, as would be done with weight decay, is not effective on its own.
> 2.  "The improvement is not significant and more competitors should be considered." We added a more recent baseline, Meta-Curvature [1], in Table 11 and Figure 4 in the Supplementary Materials.
> 3.  "The assumptions are based on the optimal predictors and thus cannot be ensured." We provide an example in Section 3.2, with the associated code in the Supplementary Material, for which the optimal predictors in the optimal representation space do not satisfy Assumption 1, while learning with the constraint on the ratio of singular values leads to a different data representation and a set of linear predictors that satisfy it. This allows to justify our regularization more rigorously and to show that in practice it may lead to significantly different empirical solutions.
>
> [1]: E. Park, J. Oliva. Meta-Curvature, NeurIPS 2019

---

### Decision · Program_Chairs · 2021-01-07
**Final Decision**

**Decision:**

Reject

**Comment:**

This paper is a systematic study of how assumptions that are present recent theoretical meta-learning bounds are satisfied in practical methods, and whether promoting these assumptions (by adding appropriate regularization terms) can improve performance of existing methods. The authors review common themes in theoretical frameworks for a meta learning setting that involves a feature learning step, based on which linear predictors for a variety of tasks are trained. Statistical guarantees for such a framework (that is, statistical guarantees for the performance of trained on an additional target task) are based on the assumption that the set of weight vectors of the linear predictors span the space (ie exhibit variety) and that the training tasks all enjoy a similar margin separability (that is, that the representation is not significantly better suited for some of the tasks than others).

The current submission, cleanly reviews the existing literature, distills out these two properties and then proposes a regularization framework (that could be added to various meta-learning algorithms) to promote these properties in the learned feature representation.

Finally, the authors experimentally evaluate to what degree the properties are already observed by some meta learning methods, and whether the proposed additions will improve performance. It is established that adding the regularization terms improves performance on most tasks. The authors thus argue that incorporating insights obtained form recent theoretical frameworks of analysis, can lead to improved performance in practice. Naturally, the purpose of the presented results is not to establish a new state of the art on a set of benchmark tasks, but to systematically study and compare the effect of adding regularization terms that will promote the properties that are desirable for a  feature representation based on statistical bounds.

I would argue that the research community should support this type of studies. The work is well presented and conducted. Most importantly, the study has a clear and general message, that will be valuable for researchers and practitioners working in on meta-learning.

However, the reviewers did not recommend publishing this type of study for ICLR. The authors are encouraged to resubmit their work to a different venue.